# Association between Sociodemographic Factors, Coverage and Offer of Health Services with Mortality Due to Oral and Oropharyngeal Cancer in Brazil: A 20-Year Analysis

**DOI:** 10.3390/ijerph192013208

**Published:** 2022-10-14

**Authors:** Márcio Vinicius de Gouveia Affonso, Igor Gonçalves Souza, Emerson Souza de Rocha, Eny Maria Goloni-Bertollo, Fabiana de Campos Gomes, Liliane Silva do Nascimento, João Simão de Melo-Neto

**Affiliations:** 1Institute of Health Sciences, Federal University of Pará (UFPA), Belem 66075-110, Brazil; 2Faculty of Medicine of São José do Rio Preto (FAMERP), Sao Jose do Rio Preto 15090-000, Brazil; 3School of Physiotherapy and Occupational Therapy, Federal University of Pará (UFPA), Street Augusto Corrêa, 01, Belem 66075-110, Brazil

**Keywords:** oral cancer, mouth neoplasms, oropharyngeal neoplasms, mortality, survival analysis, public health

## Abstract

To investigate the association between sociodemographic factors and variables related to oral health services in oral and oropharyngeal cancer mortality in Brazil, between 2000 and 2019. This study had an ecological design. Standardized mortality rates were compared between age group, sex, and regions. Age–Period–Cohort analysis was applied. Oral health services variables were analyzed in correlation tests. Survival analysis included Kaplan–Meier estimators, log-rank tests, and Cox regression. The mortality rate increased with age and was higher in men. Southeast and south regions had the highest rates for men, and the northeast and southeast had it for women. Age–Period–Cohort analysis showed a slight increase in female deaths and an increasing trend in the annual percent change in mortality for men over age 55. In survival analysis, males, Black individuals and southern residents were more strongly associated with death. The correlation between oral health teams’ coverage was high and negative, while the number of dental specialty centers and soft tissue biopsies had a high and positive correlation. Mortality and survival patterns were dependent on sex, age, geographic region and race/ethnicity. It was observed that preventive and diagnostic procedures were not being performed, which may be related to the increase in mortality.

## 1. Introduction

Oral and oropharyngeal cancers can affect the oral cavity (including lip, tongue, and mouth), salivary glands, and oropharynx [1,2]. For all sites of cancer cited, in 2020, an estimated 529,708 new cases were diagnosed worldwide, with 248,678 deaths [3]. According to the International Agency for Research on Cancer, approximately 642,000 new cases and 309,000 deaths are estimated in 2030, indicating that this neoplasm is a global public health problem [4].

In early 2000, Latin America and the Caribbean continent had intermediate incidence rates, and Brazil had the highest mortality rate [5,6]. However, in 2020, the estimated age-standardized incidence and mortality rates for Brazil were 6.2, and 2.9, respectively, for both sexes and per 100,000 inhabitants. It was the second country with the highest rates, after Cuba [4]. Regarding sex, men from these countries had the highest age-standardized incidence and mortality rates [7]. Moreover, the mortality rate in Brazil has increased since the 1980s [8].

This disease has multifactorial etiology, with intrinsic and extrinsic factors [9]. Age and sex are considered risk factors [10]; therefore, oral cancer is more common in men than in women, and the majority of patients are aged 50 years or older [2,10]. As extrinsic risk factors, chemical substances (primarily tobacco—either smoke or smokeless—and alcohol) and biological agents (exposure to human papillomavirus and immunosuppression) have been identified as carcinogenic factors [11]. Considering the effectiveness of public policies focused on risk factors, it is important to understand whether there have been changes over the years. 

Furthermore, social determinants of health, such as access to health information, and access and use of health services, can interfere with early diagnosis and treatment, thereafter in neoplasm-associated survival, incidence, and mortality [12,13]. As Brazil is a country of continental dimensions, with disparities in sociodemographic conditions and health indicators [14], this study aimed to investigate the association between sociodemographic factors and variables related to the offer and coverage of oral health services in oral and oropharyngeal cancer mortality and survival in Brazil.

## 2. Materials and Methods

The database used in this research contains consolidated information, without identifying individuals, therefore, according to National Health Council Resolution No. 510, 7 April 2016, evaluation by the research ethics committee was not required [15].

This study had a retrospective ecological design and examined secondary data of individuals who had died due to oral and oropharyngeal cancer in Brazil between 2000 and 2019. Oral and oropharyngeal cancer was defined according to the 10th revision of the International Classification of Diseases (ICD-10) by codes C00 to C06 (oral cavity) and C09–C10 (oropharynx) [1]. Cases diagnosed prior to the 2016 World Health Organization classification were not reclassified.

The units of analysis investigated were the five geographic regions of Brazil, according to the Brazilian Institute of Geography and Statistics. To illustrate those Brazilian regions, a map was constructed with QGIS Desktop 3.10.14 software and it can be seen in Figure 1. The sociodemographic variables and those related to the provision and coverage of oral health services were obtained from the Department of Informatics of the Unified Health System [16], which is an open-access database, maintained by the Brazilian Ministry of Health, and consolidates information from the National Registry of Health Establishments, the Outpatient Information System (SIA), and the Mortality Information System. 

The Mortality Information System is a national epidemiological surveillance system, created in 1975, aimed to consolidate data regarding death registries in all Brazilian cities. The death certificate is the registry document used to certify the cause of death, and it is standardized in all national territories [17,18]. According to the World Health Organization, the vital registration data quality of Brazil is high, with a completeness of 100% and usability of around 73 to 84% [19]. Additionally, the percentage of deaths coded as ill-defined causes in Brazil was 10.14 in 2015 [20].

The population quantity used to calculate the specific crude mortality rate (CMR) was based on the projection made by the Brazilian Institute of Geography and Statistics [21]. To calculate the age-standardized mortality rate (ASMR), the world standard population according to the World Health Organization was used [22].

The independent variables were geographic regions (north, northeast, south, southeast, and midwest), sex (female and male), age (≤29 years, 30–39 years, 40–49 years, 50–59 years, 60–69 years, 70–79 years, and ≥80 years), and race/ethnicity according to skin color: White, Yellow, Brown, Black, and Indigenous. The Brazilian Institute of Geography and Statistics classification states that race/ethnicity is verified by self-declaration. Furthermore, the differences between groups in ethnic, linguistic, cultural, or historical characteristics were not well established, thus any attempt to classify people according to other categories such as Caucasian or Hispanic, in Brazil, could be inaccurate. However, it was not possible to calculate the CMR considering race/ethnicity, since the population projection for this variable was not available. Thus, we considered this data only for the survival analysis, where non-informed data of race/ethnicity representing 5.16% of the total deaths were excluded. The number of deaths used to calculate the CMR was obtained according to the place of residence. Initially, the CMR was standardized by age. Then, the ASMR was standardized by sex. Both standardizations were made using the indirect method compared to the world standard population, per 100,000 inhabitants.

To analyze the correlation between oral cancer mortality rates and indicators associated with coverage and offer of oral health services, some independent variables were selected in primary health care and specialized health care. The oral health team’s coverage represents the proportion of the population assisted by oral health professionals that integrate an oral health team and are responsible for a set of actions in primary health care, such as health promotion, disease prevention, diagnosis, and treatment [23,24]. The number of first programmatic dental consultations, which is a procedure executed in primary health care aimed at the evaluation of general health conditions and dental clinical examination for diagnostic purposes. Regarding specialized health care, the chosen variables included the number of dental specialty centers where specialized dental services are offered, with emphasis on the diagnosis and detection of oral cancer [25,26], as well as the number of soft tissue biopsies of the mouth performed for diagnostic purposes. All these data were transformed into specific rates per 100,000 inhabitants for statistical analysis. However, as the oral health services variables in primary and specialized health care were available only from the period between 2011−2018, the correlation was limited to this time interval.

### 2.1. Comparison Analysis

Descriptive analysis for CMR and ASMR were expressed as measures of central tendency (mean (parametric) or median (non-parametric)), dispersion (standard deviation (parametric) or maximum, minimum, and interquartile (non-parametric)), and graphics (column bars (parametric) or box plot (non-parametric)). To verify the normality, data were submitted to the D’Agostino and Pearson test. To determine whether there were statistically significant differences between the means of ASMR in age groups and geographic regions, one-way ANOVA (parametric) and Kruskal–Wallis (non-parametric) tests were used. As post hoc tests, Tukey and Dunn’s were used in one-way ANOVA and Kruskal–Wallis, respectively. To compare ASMR according to sex, unpaired *t*-test was selected. Additionally, two-way ANOVA with Sidak’s post hoc tests was applied to verify if there were differences in mortality rates according to geographic regions and ICD code distribution. 

### 2.2. Association Analysis

To evaluate the association between mortality rate and the health services variables, Pearson’s correlation coefficient (r) was calculated to analyze the parametric data, while Spearman’s correlation (rs) was selected when the distribution was non-parametric. Pearson’s and Spearman’s were classified as negligible (≤0.30), low (0.31–0.50), moderate (0.51–0.70), high (0.71–0.90) and very high (>0.91) [27].

### 2.3. Age–Period–Cohort (APC) Analysis

To identify the influence of age, period of death, and cohort of birth on deaths due to oral and oropharyngeal cancer, the Age–Period–Cohort (APC) Web Tool (Biostatistics Branch, National Cancer Institute, Bethesda, MD, USA) [28] was applied. This web tool provides important APC functions that are useful in cancer applications: Cross-sectional age curve (cross-age) and longitudinal age curve (long-age) indicate the age effects; the fitted temporal trends and period rate ratios (PeriodRR) demonstrate the period effects; finally, cohort rate ratios (CohortRR) and local drifts represent the cohort effects. Furthermore, Wald tests followed by chi-square (χ^2^) were conducted to identify statistically significant variables according to age, period, and cohort factors. By default, the APC web tool uses the median age and period range as reference points for calculations. These reference values are calculated by the following formulas: reference age = (number of age groups + 1)/2; reference period = (number of periods + 1)/2, and reference cohort = (reference period − reference age + number of age groups).

In addition, the annual percentage change of the expected age-adjusted rates over time, which is represented by the net drift, was also calculated. *p*-values < 0.05 were considered statistically significant. Each of these functions was defined in an article by Rosenberg, Check, and Anderson [28]. Since the percentage of deaths in individuals aged less than 24 years was very low (0.22% in men and 0.59% in women), the APC analysis was applied only considering deaths above 25 years. 

### 2.4. Survival Analysis

For survival analysis, we used the package microdatasus [29] to obtain the survival time considering the period from birth until death according to sex, race/ethnicity, and geographic region. The Kaplan–Meier estimator was used to construct survival curves, then, the log-rank test was conducted to compare these curves. The variables that presented a statistically significant *p*-value were included in a Cox’s proportional hazards model for survival time to investigate if there was an association between the survival time and the different variables categories. Then, univariate and multivariate models were obtained. The data violated the proportionality assumption of the hazard ratio, but we considered that the *p*-value is dependent on the sample size, and a large sample size will produce a high significance with a minimal violation of the assumption, as reported by In and Lee [30].

SPSS Version 26.0 (IBM Corp. Released 2019. IBM SPSS Statistics for Windows, Version 26.0. IBM Corp., Armonk, NY, USA) and RStudio Team (2021). RStudio: Integrated Development Environment for R. was used for the statistical analyses.

## 3. Results

We identified 92,243 deaths due to oral and oropharyngeal cancer between 2000−2019 in Brazil, and males represented 79.63% of this total. A descriptive analysis of the entire cohort can be seen in Table 1.

In Brazil and for both sexes combined, the average CMR, per 100,000 inhabitants, was 2.36. In 2000, the CMR was 1.82 and increased to 2.95 in 2019 (Figure 2A). In females the overall mortality was 0.94 (Figure 2B) while for males it was 3.85 (Figure 2C).

After adjustment, the ASMR varied from 2.44, in 2000, to 2.57 in 2019. Males presented an ASMR of 4.26 in 2000 and an increase of 4.43% in 2019. Females showed an ASMR of 0.84 in 2000 and 0.98 in 2019, an increase of 16.75%.

### 3.1. Comparison Analysis

The mean ASMR was higher in men (4.39 ± 0.11/100,000 inhabitants) than in women (0.92 ± 0.05/100,000 inhabitants) (*p* < 0.0001, unpaired *t*-test). The two-way ANOVA analysis indicated that there was a significant difference in the ASMR according to regions (*p* < 0.0001), and ICD codes (*p* < 0.0001). Additionally, the interaction between those two independent variables presented a significant effect (*p* < 0.0001), as can be seen in Figure 3A,B.

The female-specific ASMR observed in the northeast and southeast were the highest (Figure 4A). For males, the southeast and south regions had the highest ASMR (Figure 4B). According to the age group, the mortality trend was the same in both sexes, with a progressive increase according to age (Figure 5A,B).

### 3.2. Age–Period–Cohort Analysis

The APC analysis showed that age and period factors had an influence on the CMR in both sexes, but only in men was observed the cohort effect (Figure 6A and Figure 7A). The long vs. cross RR graph represents the net drift, where we can see an increased tendency of the RR for women over time (Figure 6B), while in men there was a decrease (Figure 7B). Regarding the period effects on mortality, after the year 2007 (reference period), there was a slightly increased tendency for women (Figure 6C,D), while a decrease was observed in men (Figure 7C,D). For the cohort effects, in comparison to 1955 (reference cohort), men born between 1920–1950, and 1960–1985 presented lower RRs (Figure 6E and Figure 7E). Regarding the local drifts, we could conclude that the estimated annual percentage change of the CMR increased for men above 55 years and decreased among those aged less than 50 years (Figure 6F and Figure 7F).

### 3.3. Survival Analysis

Kaplan–Meier estimates and log-rank tests indicated that the survival rate was poorer for males than females, and the median survival time was 60 years for men, and 70 years for women (Figure 8A). Regarding race, Black individuals had the poorest survival rate, with a median survival time of 59 years (Figure 8B). For geographic regions, the lowest median survival time was 60 years in the midwest (Figure 8C). Cox regression indicated that men were more associated with death (aHR 1.80; CI 1.77–1.83) than women. For Black and Brown individuals, the adjusted hazard ratio was 1.35 (CI 1.32–1.38) and 1.27 (CI 1.25–1.29), respectively, in comparison to White people. The south, southeast, and midwest showed a higher association with death, in comparison to the north region (Figure 8D).

### 3.4. Association Analysis

In relation to the ASMR, it was observed that oral health teams’ coverage presented a high and negative correlation (r = −0.86; *p* = 0.006), which means that the coverage decreased while the ASMR increased. Dental specialty centers (r = 0.81; *p* = 0.014) and the number of soft tissue biopsies (r = 0.75; *p* = 0.031) showed also a high, but positive correlation, all the variables increased at the same time (Figure 9).

## 4. Discussion

Our findings indicated that mortality and survival patterns of individuals with oral and oropharyngeal cancer were associated with sociodemographic characteristics such as sex, age, geographic region, and race in the Brazilian population. The ASMR exhibited a high negative correlation in relation to oral health teams’ coverage, which suggests that prevention of oral and oropharyngeal cancer could be affected. 

For both sexes, the northern region had the lowest ASMR. Higher rates in southeast and south regions had already been reported [6,31,32,33]. Perea et al. [31] exhibited a trend of increased incidence and mortality from mouth and oropharynx cancer from 2002–2013 in the northeast, as we found that the annual average ASMR in women in this region was one of the highest. The implementation and/or improvement in notification systems, associated with a better organization of health services during the study period have been identified as possible causes of these findings [31]. Additionally, the unequal expansion of access to health services cannot be disregarded, since it can influence these disparities [31,34,35]. Although the highest coefficients were associated with males, current studies show that the male:female ratio has been changing, with a downward trend in variation [31,33,34]. The change in risk habit prevalence could be associated with this different epidemiological pattern. The incidence rate must be considered as another possible cause; hence, where incidence is high, the mortality should be higher too. 

Regarding age groups, the highest annual average CMR belonged to individuals over 50 years, for both sexes, a pattern found in Uruguay [36] and also in Brazil [37]. Rocha et al. [34] showed that the proportion of adults over 60 years in Brazilian municipalities was associated with the mortality rate from 2002–2012. The same pattern was observed in a southeast region city, where the highest mortality rate was observed among men over 60 years [38]. Separately, men over 80 years had the highest CMR (24.99/100,000 inhabitants), while in women, this rate was 14.25/100,000 inhabitants. These findings showed that the risk of dying from cancer increased with age [33], influenced by aging in the process of carcinogenesis and tumor growth, mainly due to immunosenescence, which leads to impairment of regenerative capacity and tissue repair [39,40].

An APC analysis of the Brazilian population [41] with death data from 1983–2017 showed that the risk of mortality increased from 40 years of age in men and from 55 years in women. Our findings suggest an increase in female deaths due to oral and oropharyngeal cancer, and also an increase in the estimated annual percentage change of mortality in men. Yet the increase in women did not reach the higher rates observed in men; this tendency could be justified by the exposure of women to smoking in more recent decades, as well as the exposure to Human Papillomavirus, mainly due to its relationship with oropharyngeal cancer [41].

The survival rate was poorer for men, and black and brown individuals, as had already been identified in other countries [42,43]. Furthermore, these individuals were more associated with death. Shiboski identified disparities between race and sexes combined; White women exhibited better survival than Black and White men, while Black women had poorer survival than White men [43]. Some studies conducted in southeast and northeast regions’ capitals did not identify differences in survival between sexes [12,44,45]. Clinical factors such as the site of injury, histological type, staging of the disease, type of treatment, the time between diagnosis and initiation of treatment, nutritional status, and human papillomavirus exposure, also influence the survival of individuals with oral cancer [12,13,44,46,47] and may justify these results. Additionally, a study that investigated racial disparities in the use of Brazilian public dental services found that Blacks and Browns depend more on these services than do Whites [48].

Differences in survival rates between regions were observed, and the midwest presented the poorest survival. When comparing countries, differences are commonly justified by the stage of the disease in diagnosis and the availability and quality of treatment [49]. In Brazil, the unequal expansion of health services resulted in difficulties in the access and use of these services [38,43]; therefore, the access to diagnosis and treatment in different regions and localities, as for some specific groups, could get worse [6] and could be associated with our finding. 

The National Oral Health Policy states [26] that prevention and early detection of oral cancer can be attributed to the oral health teams in primary health care. However, oral health services still face difficulties in the integration of care [48], and delay in diagnosis is a major concern in oral cancer mortality, responsible for at least 50% of the diagnoses in advanced stages [50,51]. Regarding the oral health team’s coverage correlation, we could suggest that actions toward prevention and early detection were not being performed, [34] and this interferes in diagnosis, referral, treatment, and consequently, death. Once dental specialty centers in specialized health care are responsible for diagnosis and referral for hospital treatment [26], and we found a high and positive correlation, we could also suggest that despite the increase in the number of biopsies, the diagnosis was made in advanced stages or the referral took a long time. To corroborate this hypothesis, the number of soft tissue biopsies also presented a high and positive correlation. 

In order to plan, develop and also evaluate public health policies, especially those aimed at oral and oropharyngeal cancer, it is essential to understand the association between sociodemographic factors and mortality due to this neoplasm. However, our study has limitations that include the weaknesses in the process of inserting information into Brazilian databases, once this country has connectivity infrastructure disparities all over regions, where data can be lost or unreported until it arrives at the Ministry of Health. Furthermore, a large amount of unknown or unreported data affects the reliability of the analysis, although this situation has been improving over the years in Brazil. During the study period, 2000–2019, the percentage of deaths from symptoms, signs, and abnormal clinical and laboratory findings, not elsewhere classified (ICD-10: R00–R99) was 8.1% of the total deaths registered in the Mortality Information System. Additionally, the ecological design of this research can be defined as a limitation, since there are no individual analyzes that could be required in a more accurate investigation, as occurs in control or cohort case studies. Therefore, studies with other methodological approaches should be carried out to clarify the possible issues raised here.

## 5. Conclusions

In this study, we found that the mortality rate from oral and oropharyngeal cancer increased during the aging process, being more prevalent in men, who had a lower survival rate, regardless of the geographical region of Brazil. However, in recent decades, the mortality in women has been slightly increasing. Individuals of the Black race/skin color had the poorest survival rate. Midwest residents showed a poorer survival rate than other regions. Regarding the oral health services indicators, it was observed that preventive and diagnostic procedures were not being performed, which may be associated with an increase in the mortality rate in Brazil.

## Figures and Tables

**Figure 1 ijerph-19-13208-f001:**
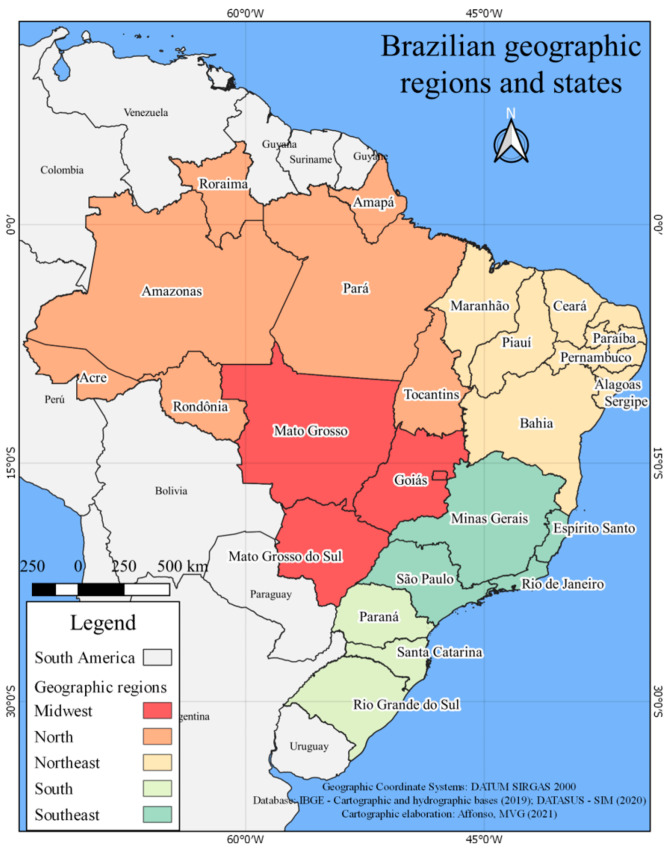
Brazilian geographic region and states.

**Figure 2 ijerph-19-13208-f002:**
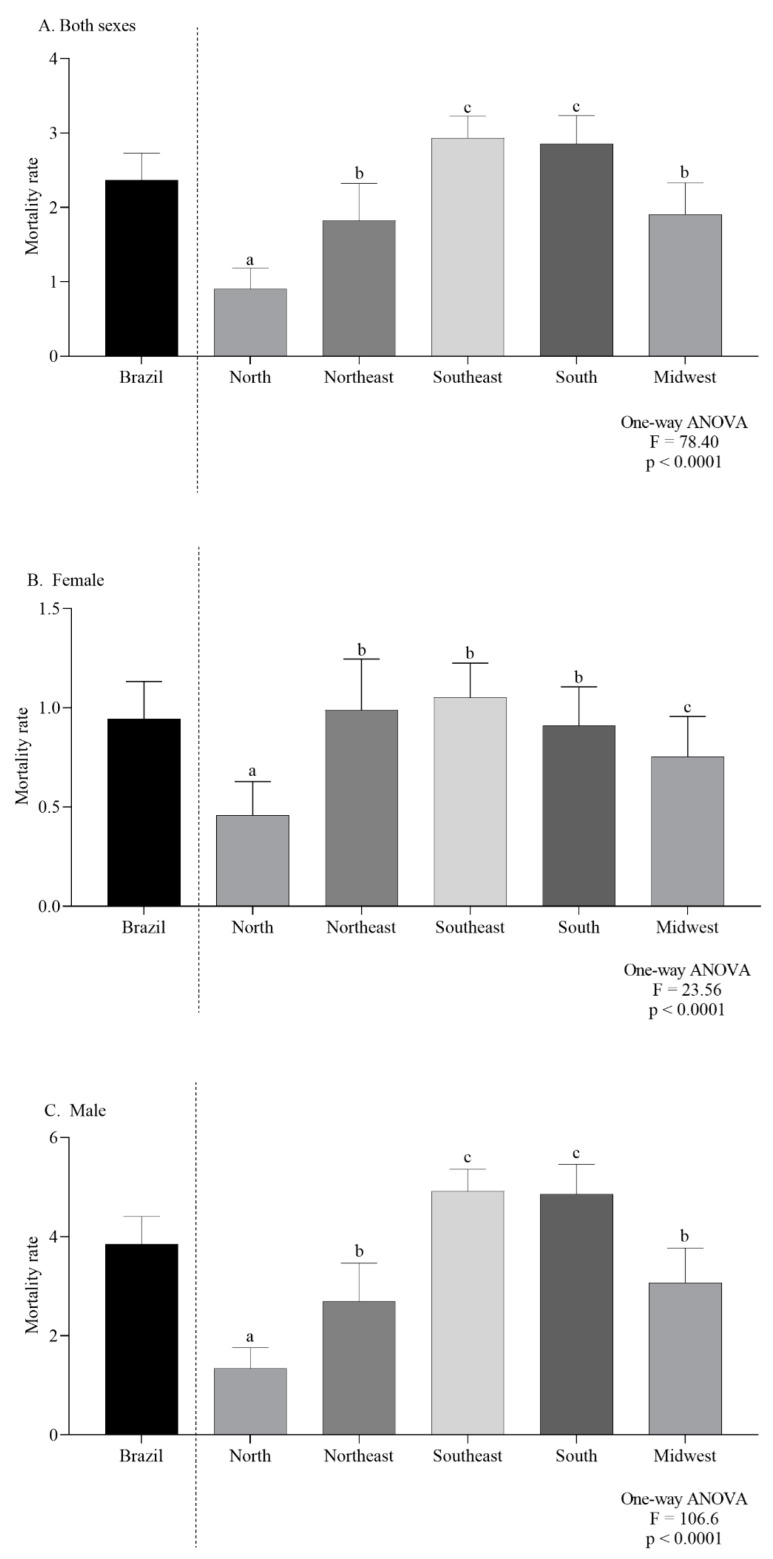
Overall mortality in Brazil, by regions, in Both sexes (**A**), females (**B**) and males (**C**). Lowercase letters indicate significant differences between regions (*p* < 0.05, Dunn’s test).

**Figure 3 ijerph-19-13208-f003:**
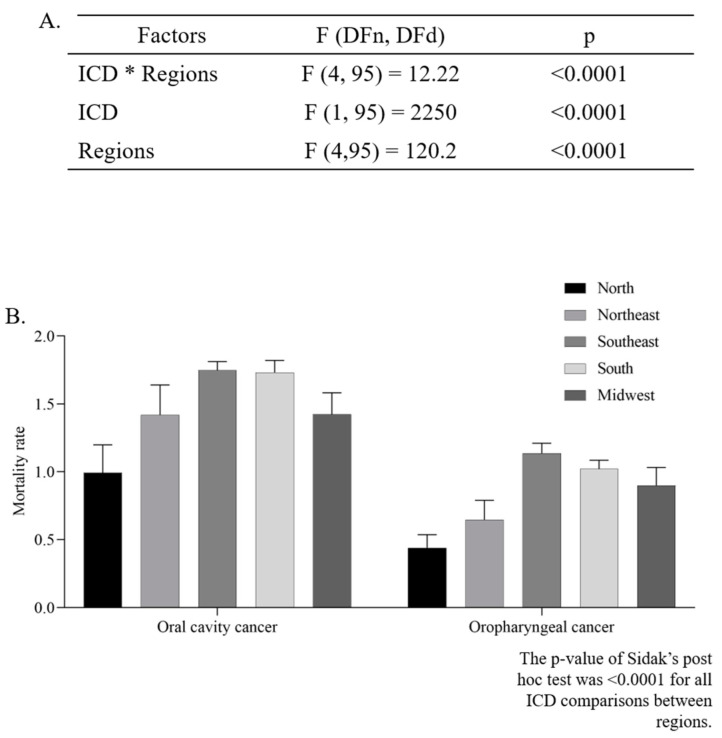
Two-way ANOVA (**A**) with average estimated crude mortality rate for every 100,000 inhabitants according to geographic region and International Classification of Diseases (ICD-10) (**B**).

**Figure 4 ijerph-19-13208-f004:**
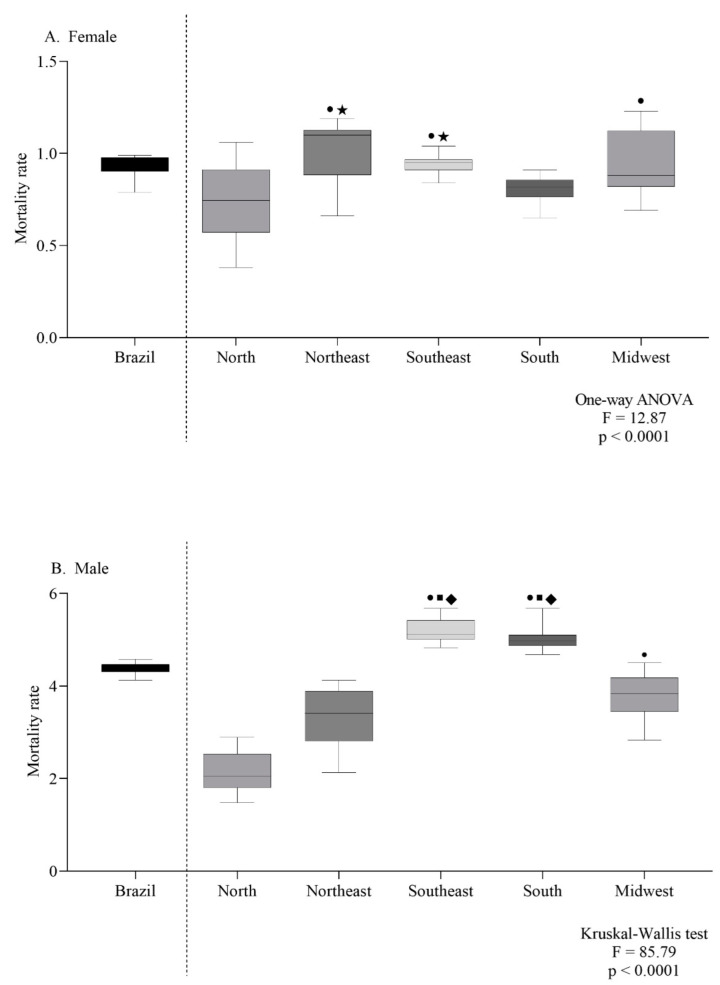
Mortality rate for every 100,000 inhabitants per sex (female (**A**) or male (**B**)) according to geographic region (**A**,**B**). Symbols indicate significant differences between regions: north ●; northeast ■; midwest ◆; south ★ (*p* < 0.05, Dunn’s test).

**Figure 5 ijerph-19-13208-f005:**
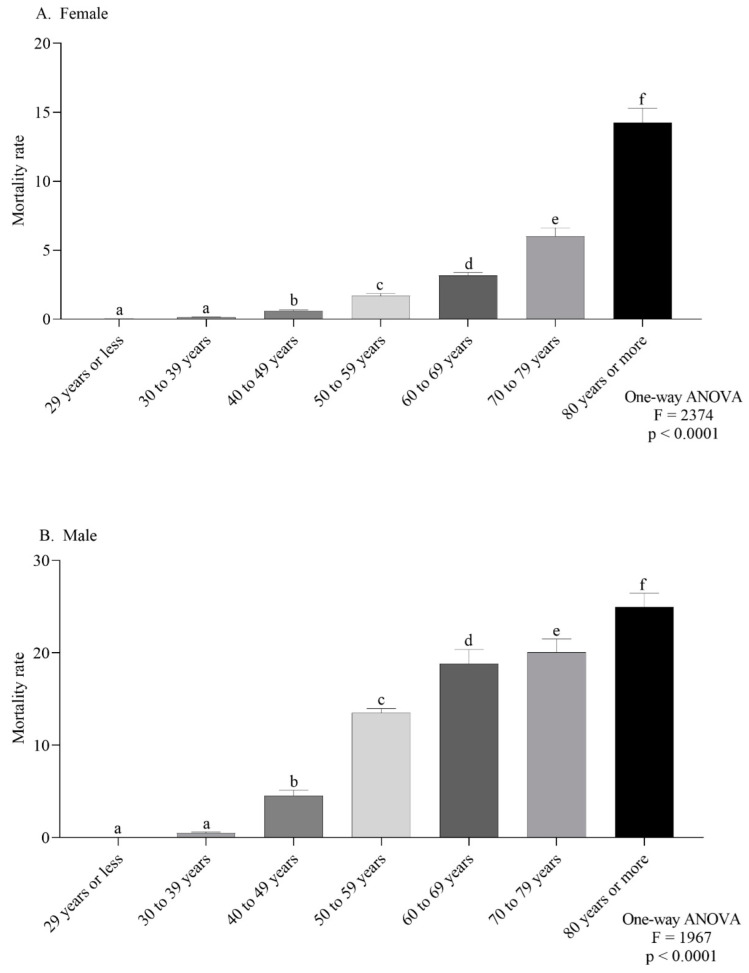
Mortality rate for every 100,000 inhabitants per sex (female (**A**) or male (**B**)) according to age group (**A**,**B**). Lowercase letters indicate significant differences between age groups (*p* < 0.05, Dunn’s test).

**Figure 6 ijerph-19-13208-f006:**
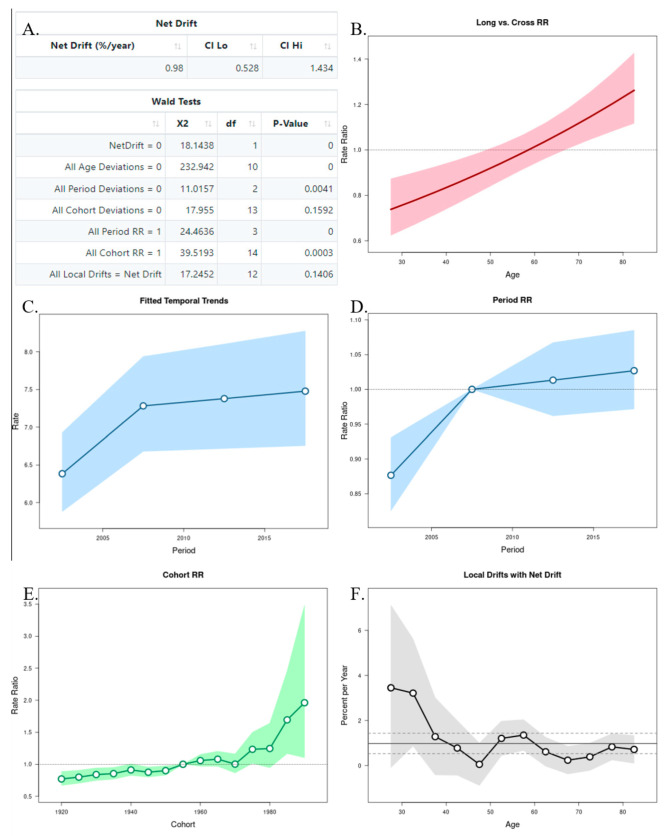
Age–Period–Cohort (APC) analysis with Wald test (**A**), representation of the age (**B**), period (**C**,**D**), and cohort effects (**E**,**F**) in females.

**Figure 7 ijerph-19-13208-f007:**
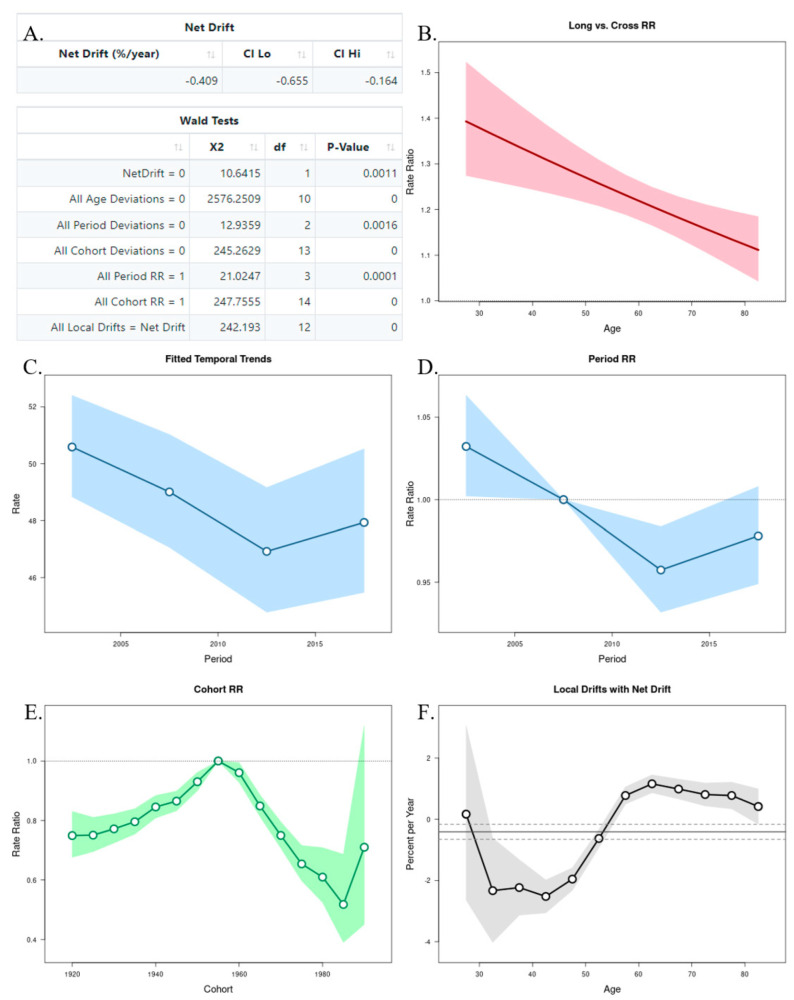
Age–Period–Cohort (APC) analysis with Wald test (**A**), representation of the age (**B**), period (**C**,**D**), and cohort effects (**E**,**F**) in males.

**Figure 8 ijerph-19-13208-f008:**
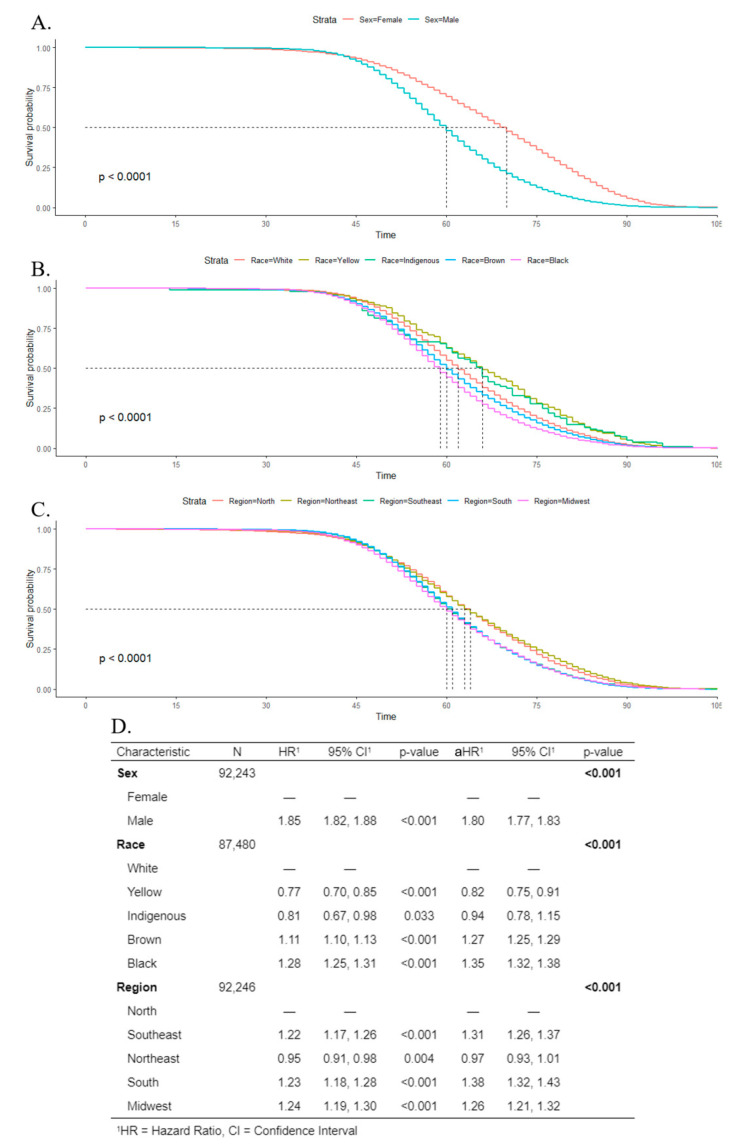
Kaplan–Meier survival rates for individuals with oral cancer according to sex (**A**), race (**B**), and geographic regions (**C**). Cox’s proportional hazards model for survival time with univariate (hazard ratio (HR)) and multivariate (adjusted hazard ratio (aHR)) models (**D**).

**Figure 9 ijerph-19-13208-f009:**
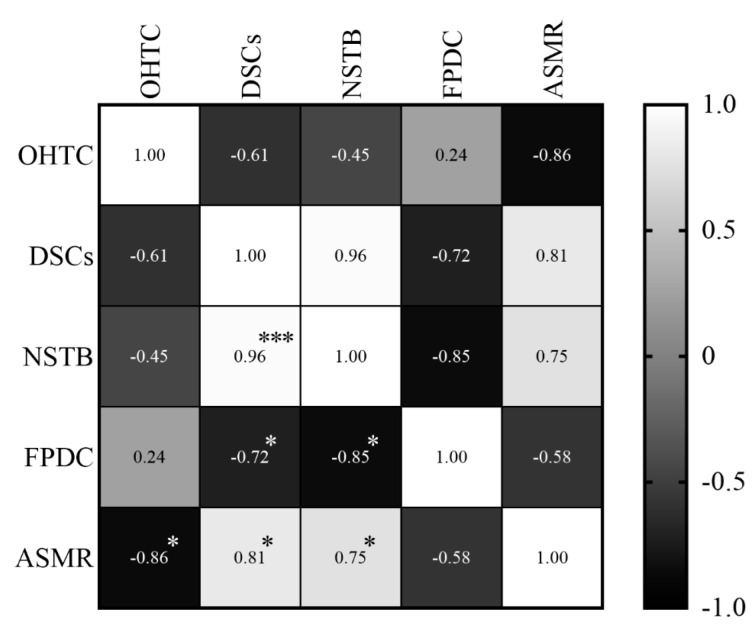
Correlations between the specific mortality rate in individuals and variables of primary and specialized care, between 2011 and 2018. The shade gradient bar shows correlation coefficient. The asterisk indicates *p*-values < 0.05 * and <0.0001 ***. Age-standardized mortality rate (ASMR), oral health teams coverage (OHTC), first programmatic dental consultations (FPDC), dental specialty centers (DSCs), and number of soft tissue biopsies (NSTB).

**Table 1 ijerph-19-13208-t001:** Descriptive analysis of the entire cohort.

	Brazilian Geographic Regions					
Variables	Brazil N = 92,243	Nort HN = 2964	Southeast N = 48,074	Northeast N = 19,739	Sout HN = 15,972	Midwest N = 5497
Sex						
Female	18,779 (20%)	743 (25%)	8853 (18%)	5490 (28%)	2599 (16%)	1094 (20%)
Male	73,464 (80%)	2221 (75%)	39,218 (82%)	14,249 (72%)	13,373 (84%)	4403 (80%)
Unknown	3	0	3	0	0	0
Race						
White	50,883 (58%)	646 (22%)	29,255 (64%)	4974 (27%)	13,591 (88%)	2417 (46%)
Yellow	397 (0.5%)	10 (0.3%)	255 (0.6%)	75 (0.4%)	39 (0.3%)	18 (0.3%)
Indigenous	101 (0.1%)	23 (0.8%)	20 (<0.1%)	32 (0.2%)	12 (<0.1%)	14 (0.3%)
Brown	28,399 (32%)	2027 (70%)	11,486 (25%)	11,424 (63%)	1040 (6.7%)	2422 (46%)
Black	7700 (8.8%)	176 (6.1%)	4565 (10%)	1770 (9.7%)	772 (5.0%)	417 (7.9%)
Unknown	4766	82	2493	1464	518	209
Age						
≤29	486 (0.5%)	44 (1.5%)	194 (0.4%)	153 (0.8%)	58 (0.4%)	37 (0.7%)
30 to 39	1919 (2.1%)	76 (2.6%)	910 (1.9%)	499 (2.5%)	297 (1.9%)	137 (2.5%)
40 to 49	12,170 (13%)	344 (12%)	6481 (13%)	2388 (12%)	2122 (13%)	835 (15%)
50 to 59	26,743 (29%)	688 (23%)	14,771 (31%)	4769 (24%)	4864 (30%)	1651 (30%)
60 to 69	24,554 (27%)	768 (26%)	13,192 (27%)	4731 (24%)	4466 (28%)	1397 (25%)
70 to 79	15,791 (17%)	608 (21%)	7826 (16%)	3785 (19%)	2646 (17%)	926 (17%)
≥80	10,574 (11%)	435 (15%)	4697 (9.8%)	3409 (17%)	1519 (9.5%)	514 (9.4%)
Unknown	9	1	3	5	0	0

## Data Availability

All data were made available by the http://tabnet.datasus.gov.br/cgi/deftohtm.exe?sim/cnv/obt10uf.def (accessed on 30 November 2020).

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
