# Peer review of "Association between Sociodemographic Factors, Coverage and Offer of Health Services with Mortality Due to Oral and Oropharyngeal Cancer in Brazil: A 20-Year Analysis"

_ijerph, 2022, doi:10.3390/ijerph192013208_

Round 1

Reviewer 1 Report

This an interesting study one mortality rates of oral and oropharygeal cancers in Brazil. However, these data has to put into perspective of all deaths in Brazil and the various regions before any conclusions can be drawn with related to sociodemographic factors etc. Now we see an effect related to, a.o., age and region, but the mortality in general can also be higher in advanced age as well as in these regions. Moreover, In the figures, what represents the data, are it means and SD, or medians with IRQs and min/max, or something else. This is not explained for which I can not understand the figures properly.

Author Response

Response to Reviewer 1 Comments

Point 1: This an interesting study one mortality rates of oral and oropharygeal cancers in Brazil. However, these data has to put into perspective of all deaths in Brazil and the various regions before any conclusions can be drawn with related to sociodemographic factors etc. Now we see an effect related to, a.o., age and region, but the mortality in general can also be higher in advanced age as well as in these regions.

Response 1: Thank you for your comment. To ascertain the effects of sociodemographic factors on oral and oropharyngeal cancer mortality in Brazil, we initially identified the number of deaths at the national level; subsequently, the crude mortality rate was calculated for each geographic region. Then, these crude rates were adjusted by the standard population established by the World Health Organization, using the indirect method. By adjusting mortality by age groups, we allowed the comparison between regions and the identification of the pattern of mortality, according to some sociodemographic factors. Please see the Materials and Methods section.

Point 2: Moreover, In the figures, what represents the data, are it means and SD, or medians with IRQs and min/max, or something else. This is not explained for which I can not understand the figures properly.

Response 2: Thank you for your comment. The bar figures represent the parametric numerical data, therefore, the variations presented in these images refer to the standard deviation, while in the boxplot figures, the medians, IRQs and min/max are represented. The description was further detailed in the “Comparison analysis” section. Please see the highlighted text (page 4, lines 132-135).

Reviewer 2 Report

I read with interest the manuscript entitled: "Association between sociodemographic factors, coverage and offer of health services with mortality due to oral and oropharyngeal cancer in Brazil: A 20-year analysis".

The manuscript is well structured. I consider the research to be of interest and up to date on the topic.

The authors should respond to the following points:

- The authors should improve the resolution of figures number 2, 5, 6 and 7.

- What are the limitations of the study?

- The bibliographical references are not described according to the journal's regulations.

Author Response

Response to Reviewer 2 Comments

Point 1: The authors should improve the resolution of figures number 2, 5, 6 and 7.

Response 1: Thank you for your comment. As requested, the figures' resolution were adjusted to 300dpi.

Point 2: What are the limitations of the study?

Response 2: Thank you for your comment. The limitations of the study are described in the second sentence of the last paragraph of the discussion section. Please see pages 14-15, lines 347-359.

Point 3: The bibliographical references are not described according to the journal's regulations.

Response 3: Thank you for your comment. As requested, the references were adjusted according to the journal's regulations.

Reviewer 3 Report

Methods and Results: because there are so many analyses being run in this study, I suggest using subheading under each of methods and results to help readers keep track of what tests are being run to answer what question.

Results: you've launched straight into the mortality analysis, but a descriptive snapshot (table) of the entire cohort used for analysis would be good to allow readers to see the overall spread of this group in terms of the major demographic variables. 

Pages 8+9: were 2007 and the 1955 cohort chosen as references for any particular reasons or to divide the data into two halves? 

Figures 7B and 7C: possible to make these figures larger so the different coloured lines can be visually separated? Especially Figure 7C where the different races are so close to each other.

Figure 7D: can you make it clear which columns are uni and which ones are multi models.

line 231: "...OHTC presented A high and negative..." same error in a few other places in the manuscript, please double check.

Line 235: figure 8 misquoted as figure 7?

Figure 8: can you add a legend to this figure to indicate that the shade gradient bar shows correlation coefficient and the asterix numbers for different p-values.

Lines 250-252: sentence beginning "The implementation and/or improvement in notification systems..." please add reference for this possible cause.

Line 273: "...the increase in women DID not REACH..."

Line 294: "...could GET worse..."

Given the other similar studies conducted in Brazil, can you briefly highlight how your study stands out from previous research?

Author Response

Response to Reviewer 3 Comments

Point 1: Methods and Results: because there are so many analyses being run in this study, I suggest using subheading under each of methods and results to help readers keep track of what tests are being run to answer what question.

Response 1: Thank you for your comment. As requested, subheadings were added to the methods and results sections.

Point 2: Results: you've launched straight into the mortality analysis, but a descriptive snapshot (table) of the entire cohort used for analysis would be good to allow readers to see the overall spread of this group in terms of the major demographic variables.

Response 2: Thank you for your comment. As suggested, a descriptive table was made. It can be seen in Table 1.

Point 3: Pages 8+9: were 2007 and the 1955 cohort chosen as references for any particular reasons or to divide the data into two halves?

Response 3: By default, the Age Period Cohort analysis tool uses the median age and period ranges as reference points for calculations.

The reference values are calculated by these formulas:

Reference age = (Number of age groups +1)/2

Reference period = (Number of Periods +1)/2

Reference cohort = (Reference Period – Reference age + Number of Age Groups)

To clarify this analysis, we added these sentences to the methods section. It can be seen in page 5, lines 163-167.

Point 4: Figures 7B and 7C: possible to make these figures larger so the different coloured lines can be visually separated? Especially Figure 7C where the different races are so close to each other.

Response 4: Thank you for your comment. As requested, Figure 7 was resized for better visualization.

Point 5: Figure 7D: can you make it clear which columns are uni and which ones are multi models.

Response 5: Thank you for your comment. As requested, the letter “a” for “adjusted” was added in the multimodel column, and also in the figure’ legend. It can be seen in page 13, lines 263-264.

Point 6: line 231: "...OHTC presented A high and negative..." same error in a few other places in the manuscript, please double check.

Response 6: Thank you for your comment. As requested, this error was corrected in all sentences.

Point 7: Line 235: figure 8 misquoted as figure 7?

Response 7: Thank you for your comment. As identified, the figure was misquoted. Now it is correct, as can be seen in line 265-266.

Point 8: Figure 8: can you add a legend to this figure to indicate that the shade gradient bar shows correlation coefficient and the asterix numbers for different p-values.

Response 8: Thank you for your comment. As requested, a legend was added in Figure 8. It can be seen in line 272.

Point 9: Lines 250-252: sentence beginning "The implementation and/or improvement in notification systems..." please add reference for this possible cause.

Response 9: Thank you for your comment. As requested, a reference was added and it can be seen in line 290.

Point 10: Line 273: "...the increase in women DID not REACH..."

Response 10: Thank you for your comment. As suggested, the sentence was rewritten. It can be seen in line 311.

Point 11: Line 294: "...could GET worse..."

Response 11: Thank you for your comment. As suggested, the sentence was rewritten. It can be seen in line 332.

Point 12: Given the other similar studies conducted in Brazil, can you briefly highlight how your study stands out from previous research?

Response 12: Thank you for your comment. In addition to showing the pattern of oral and oropharyngeal cancer mortality in the Brazilian population, according to some sociodemographic characteristics, our study evaluated the association between oral and oropharyngeal cancer mortality and variables related to the coverage and provision of health services. In this analysis, we identified that the performance of preventive exams and early diagnosis was impaired, which may have been contributing to the increase in the overall mortality rate.

This finding is essential for the planning and reformulation of public policies aimed at the prevention and control of oral and oropharyngeal cancers, since it identifies where the gaps are in the provision of primary health care in the Brazilian health system.

Round 2

Reviewer 1 Report

The paper has improved, but it is hard to read due to the many abbreviations used. Use only abbreviations that are often repeated, such as ASMR, not the other abbreviations.

Also the bar graphs are difficult to read. E.g., the mortality rate in females is probably normal distributed and in male non normal. Use a similar method, in case of a normal distribution than the IQRs are rather equal distributed. I will do it for all graphs.

Furthermore, what is the explanation for the observed differences, different races, different circumstances, different care etc.  This has to be explained. Add also a graph of the overall mortality rate for the different regions in females and males and the overall mortality rate from cancer.

Author Response

Point 1: The paper has improved, but it is hard to read due to the many abbreviations used. Use only abbreviations that are often repeated, such as ASMR, not the other abbreviations.

Response 1: Thank you for your comment. As required, the abbreviations were reduced to those that presented more than five times in the text, being CMR, ASMR and APC with their analyses.. The following abbreviations were taken from the text IARC, ASR, HPV, WHO, IBGE, DATASUS, CNES, SIM, PHC, SHC, OHTC, OHT, FPDC, DSCs, NSTB. However, abbreviations were kept in figures when necessary and explained in their respective captions, because they impair image quality.

Point 2: Also the bar graphs are difficult to read. E.g., the mortality rate in females is probably normal distributed and in male non normal. Use a similar method, in case of a normal distribution than the IQRs are rather equal distributed. I will do it for all graphs.

Response 2: Thank you for your comment. This comment is very valid so that we can have a flow of experiences. Although the IQRs are similar, it is appropriate to present a column chart for parametric data, and a box plot for non-parametric analysis, since the results are provided by statistical analysis with variables of central tendency and dispersion specific to each normality. However, as requested, we adapted the above mortality rate graph to a box plot.

Point 3: Furthermore, what is the explanation for the observed differences, different races, different circumstances, different care etc. This has to be explained. 

Response 3: Thank you for your comment. We pointed out possible causes to the differences observed in different sentences of the discussion section, as following: in the second paragraph we listed the factors associated to differences in the geographic regions from the line 298 to 302;  in relation to sexes, our hypothesis are described in lines 302 to 306; regarding age groups, the explanation to our findings are in lines 313 to 316; and considering the survival analysis between regions and race/color, we identified possible factors that are listed in lines 331 to 343.

Point 4: Add also a graph of the overall mortality rate for the different regions in females and males and the overall mortality rate from cancer. 

Response 4: Thank you for your comment. As suggested, a graphical representation of the overall mortality for the different regions, in both sexes, combined and separated, was made.